# Spatially distinct FRL and Ena dependent actin networks coordinate nuclear positioning in Drosophila nurse cells

Rita Gombos[1], Dávid Farkas[1], Balázs Vedelek[1], Szilárd Szikora[1], József Mihály[1,2]*

1 Institute of Genetics, HUN-REN Biological Research Centre, Szeged, Hungary, 2 Department of Genetics, University of Szeged, Szeged, Hungary

* mihaly.jozsef@brc.hu

## Abstract

Position of the nucleus is dynamically controlled to ensure a variety of cellular functions in a broad range of organisms form yeast to human. Nuclear positioning in *Drosophila* nurse cells is crucial during dumping when cells transfer their entire cytoplasmic content into the oocyte. An important prerequisite of effective dumping is the formation of an array of actin cables which holds the nucleus in a central position, thereby allowing transmission of the cytoplasmic cargo. Here we report the identification of FRL, a formin type of actin assembly factor, as a novel determinant of cytoplasmic actin bundle formation. We found that FRL and the formerly described Ena protein display a differential requirement. Comparison of the *frl* and *ena* loss of function situations revealed that FRL is mainly required for creation of the cytoplasmic actin subpopulation at stage 10B, while Ena mostly promotes formation of a ring canal attached actin array, already present at stage 7 and persists till dumping. Upon the concurrent absence of FRL and Ena the nuclear positioning actin cables are completely missing, strongly suggesting that nuclear positioning in the nurse cells requires the coordinated action of two spatially distinct actin networks.

## Author summary

Controlling the position of the nucleus in multicellular organisms is vital for life from zygote formation throughout development and the maintenance of normal homeostasis. Cells use cytoskeleton dependent forces to push or pull their nuclei into the appropriate position. In the *Drosophila* egg chambers, made up from 15 nurse cells and the oocyte, the nurse cell nuclei need to be kept away from the ring canals (cytoplasmic bridges between these germline cells) during the last stages of oogenesis to prevent clogging and allow the bulk transport of materials (dumping) towards the oocyte. This was thought to be achieved by cytoplasmic actin cables spanning from the cortical membranes to the nucleus. Whereas we

**Data availability statement:** All relevant data are within the manuscript and its Supporting information files.

**Funding:** This work was supported by the Hungarian Scientific Research Found (OTKA) (K132782 to J.M.,), by The National Laboratory of Biotechnology through the Hungarian National Research, Development and Innovation Office (NKFIH) (grant No. 2022-2.1.1-NL-2022-00008 to J. M.), and OTKA Postdoctoral Fellowship (PD 121193 to R.G.). The funders had no role in study design, data collection and analysis, decision to publish, or preparation of the manuscript.

**Competing interests:** The authors have declared that no competing interests exist.

present further evidence in support of the importance of these actin arrays, we found that the prominent cytoplasmic actin bundles represent only one part of a two-tier mechanism, critically relying on the coordinated action of two distinct actin networks that differ in both the temporal and spatial regulation of their initiation, and their contribution to dumping efficiency. We hope that clarification of an important mechanistic aspect will be imperative to gain novel insights into the general means of nuclear positioning in fruit flies, and potentially, in other organisms.

## Introduction

Controlling the position of the nucleus in multicellular organisms is vital for life from zygote formation throughout development till the maintenance of normal homeo-stasis. In the newly fertilized embryo, a delicate nuclear positioning system ensures apposition of the male and female pronucleus, which is a prerequisite of zygote for-mation (see for review [1,2]). Once differentiation takes place, a high number of cells undergo asymmetric divisions, cell shape changes, cell migration and rearrange-ments, processes that all involve proper positioning of the nucleus (see for review [2,3]). The maintenance of normal functioning of an organism or renewal of a tissue often requires similar mechanisms, such as asymmetric cell division, polarized mate-rial transport and cell migration. In addition, germ cell production is another important context in which nuclear positioning plays a critical role (see for review [4,5]). Col-lectively, these examples highlight that the regulation of nucleus position is a major factor in cellular organization and function throughout life.

Cells either use microtubule or actin dependent cytoskeletal forces to place their nucleus into the correct position during development or cell movement [1]. In the *Drosophila* ovary, nurse cells regulate the position of their nucleus via cytoplasmic actin cables during late oogenesis, which is crucial for fertility. The *Drosophila* egg chambers are comprised of 16 germline cells, surrounded by a monolayer of follicu-lar cells (Fig 1I). The 16 germline cells are derived from a single cystoblast cell that undergoes a series of four incomplete cell divisions to generate a single oocyte and 15 nurse cells, interconnected by cytoplasmic bridges called ring canals (RCs) [6]. The main function of the nurse cells is to support the oocyte by transporting various products (mRNAs, proteins, metabolites and organelles) into the oocyte through the ring canals. This transport process can be subdivided into two phases, an early phase (up to stage 10A) characterized by slow cytoplasmic streaming, and a rapid late phase (from stage 10B) when the nurse cells contract in order to dump their entire cytoplasmic contents into the oocyte [7,8]. It was shown that, during the early phase, directional flow of the cytoplasmic materials and vesicles is assisted by ring canal attached actin baskets that form at stage 6–7, and display an asymmetric mor-phology at the nurse cell-oocyte borders [9–11]. At stage 10B, just before dumping begins, a prominent cytoplasmic actin cable array arises in the nurse cells spanning from their cortical membranes to the nucleus, thereby precluding the nuclei from

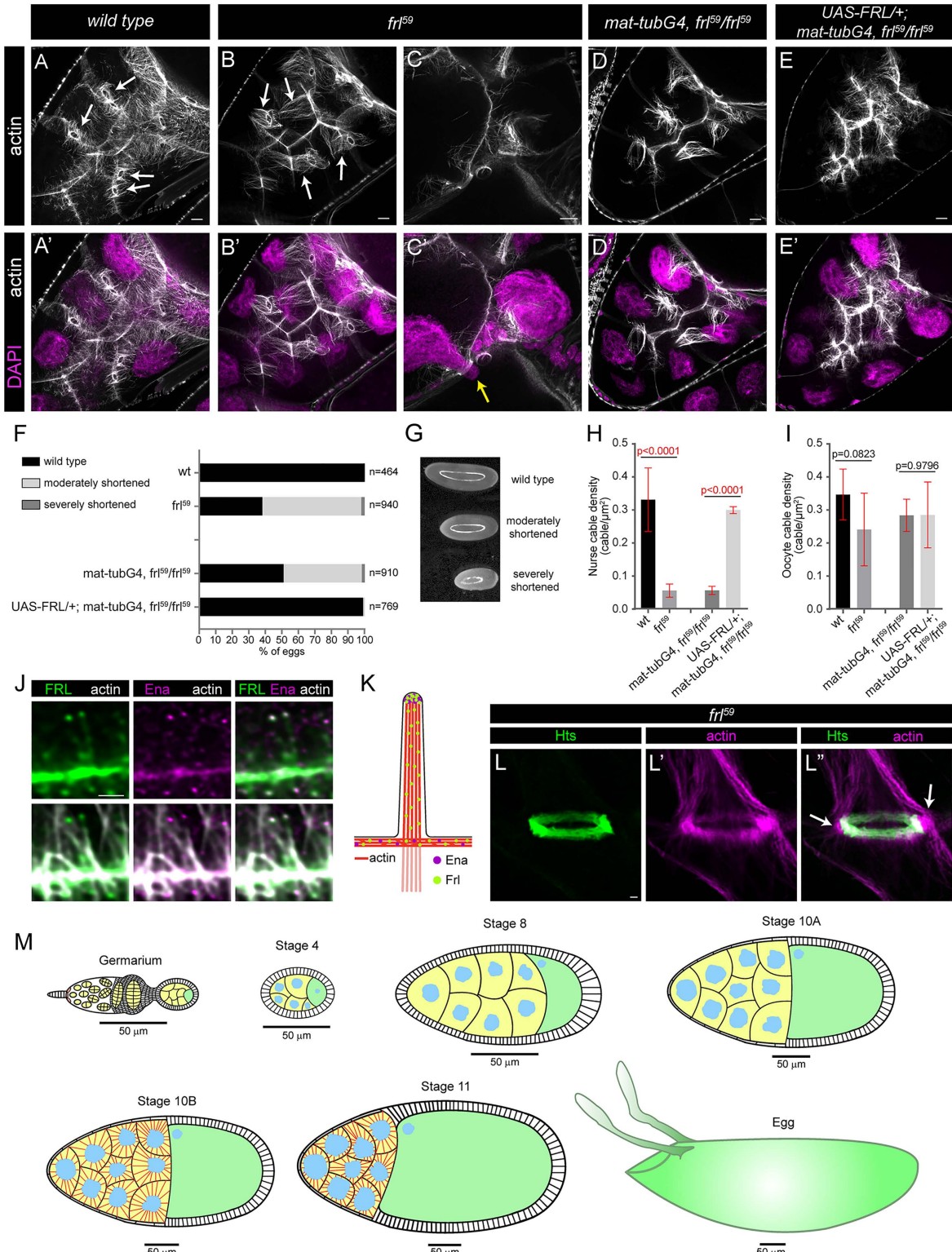

**Fig 1. FRL is required for cytoplasmic actin formation in Drosophila egg chambers.** (**A-E'**) Actin and nuclear (DAPI, in magenta) staining of stage 10B egg chambers of the indicated genotypes. (**A, A'**) In nurse cells of a wild type egg chamber the cytoplasmic and the ring canal associated (arrows in A) actin bundles are both visible. (**B-C'**) In *frl⁵⁹* mutant egg chambers the cytoplasmic cables are mostly missing, while the ring canal cables appear

longer and more prominent (arrows in B) than in wild type, and in some cases the nurse cell nucleus blocks the ring canal (yellow arrow in C'). **(D-E')** Phenotype of the *frl^59* mutants can be rescued by *maternal-tubGal4* driven expression of full length FRL (*UAS-FRL*). **(F)** Quantification of the egg length phenotype laid by mothers with the indicated genotypes. Egg length of 0,48-0,6 mm is classified as wild type, 0,38-0,48 mm as a moderately shortened, while shorter than 0,38 mm as severely shortened egg size. *n* indicates the number of eggs measured. **(G)** An illustration of the different egg length categories quantified in panel F. **(H, I)** Quantification of nurse cable (H) and oocyte cable (I) actin density in egg chambers with the genotype indicated. **(J)** The localization of FRL and Ena at the cytoplasmic actin bundles. Note the strong colocalization at the tip of the actin bundles, and a weaker FRL signal along the membrane pits, schematized in **K**. **(L-L")** A ring canal from an *frl^59* mutant egg chamber stained for Hts (in green) to label the ring canal proper, and actin (in magenta). Arrows in L" indicate peripheral areas revealing that the RC cables initiate from the vicinity of the ring canals outside of the RC proper. **(M)** Schematic drawings illustrating a few major stages of *Drosophila* oogenesis. Nurse cells are in yellow, oocyte is in green, nuclei are in blue. Note the cytoplasmic actin cables (in red) that form in stage 10B. Scale bars: 1 μm in panel G and L, 10 μm in all other panels.

clogging the ring canals. These actin filaments initiate at the plasma membrane as microvilli-like, short membrane evaginations protruding into the neighboring nurse cells. Although some pioneering studies suggested that the actin bundles are arranged into an overlapping pattern [12], later examinations revealed that the actin cables are formed by unsegmented bundles of uniformly oriented parallel actin filaments, with their (+) ends located in the plasma membrane protrusions and (-) ends found in vicinity of the nuclei [13].

Consistent with the importance of filament bundling, in the absence of Fascin, Villin or Filamin, the cytoplasmic actin cables fail to form and organize properly [14–17]. In addition to these actin bundling factors, proteins involved in filament formation, including the actin monomer binding Profilin [18], the elongation factor Enabled (Ena) [19] and the actin assembly factor Diaphanous (Dia) [20], were also linked to cytoplasmic actin filament growth and organization. However, the lack of these proteins does not typically result in a complete absence of the cytoplasmic actin cables, instead the major effect of Dia loss was reduced filament growth rate [20], whereas the removal of Ena led to decreased cable density and growth rate [19,20], but not to an entire block of filament formation. Thus, these observations indicated that additional actin regulatory factors might also play a role in nuclear positioning during *Drosophila* oogenesis.

Here, we report that the absence of FRL (a formin type of actin assembly factor) induces a severe reduction in the number of the cytoplasmic actin bundles, which is paralleled with the formation of strong, ring canal associated cables. With help of advanced imaging techniques, we revealed that, unlike previously assumed, these RC attached actin bundles remain present till stage 10B in the wild type egg chambers as well. We show that the activity of FRL is regulated by Cdc42, and that it is strongly colocalized with Ena at the (+) tip of the actin filaments. Despite their highly similar distribution pattern, we found that FRL and Ena exert a differential effect on the nurse cell actin cytoskeleton because the *frl* null mutation impairs only the cytoplasmic actin bundles, whereas the depletion of Ena primarily affects the RC actin baskets, which appear to play a more prominent role in dumping than the cortical membrane derived cytoplasmic actin arrays. Remarkably, the concomitant loss of FRL and Ena results in the complete absence of the cytoplasmic and the RC cables, and in a very strong dumpless phenotype, indicating that the FRL and Ena dependent pathways are necessary and sufficient to govern nuclear positioning in the *Drosophila* nurse cells.

## Results

### FRL is required for cytoplasmic actin assembly and nuclear positioning in the *Drosophila* nurse cells

While studying the developmental roles of the *Drosophila* formin, FRL, we noticed that mothers homozygous for *frl^59* (an *frl* null allele) [21,22] often lay eggs smaller than normal. We quantified this phenotype and found that ~60% of the eggs exhibit a weaker or stronger reduction in egg length when compared to wild type (wild type length in our conditions varies between 0,48–0,6 mm; egg length of 0,38–0,48 mm is classified as weak reduction, whereas the category shorter than 0,38 mm is considered as strong reduction) (Fig 1F, G). To determine the origin of this defect, we studied the oogenesis in *frl^59* homozygous mutant females by immunostaining the actin cytoskeleton. We revealed that egg chambers of the mutants look largely normal up to stage 10A, however, in stage 10B we detected a strong decrease in the number of the

cytoplasmic actin filaments in the nurse cells (Fig 1A–B'). The reduction of the bundles was evident in all cytoplasmic actin subpopulations, i.e., the nurse cables (derived from nurse cell-nurse cell borders), the follicle cables (derived from nurse cell-follicular cell borders) and the oocyte cables (derived from nurse cell-oocyte borders) [20] were all affected (Figs 1A–B', S1B–G), with the strongest effect on the nurse cables (Fig 1H, I). Curiously, these alterations were paralleled with the formation of prominent actin bundles connected to the ring canals (Fig 1B, L–L"), and while the overall shape and size of the ring canals remained normal, a nearly twofold increase was detected in level of the Hts protein (S2A–B', F, G Fig). Despite looking thicker and longer, these actin structures highly resembled the ring canal attached actin baskets that normally form during stage 6–7 [9–11], and are thought to help directional transport of materials through the ring canals during the early phase of dumping. The obvious structural similarity prompted us to ask whether these actin baskets persist till stage 10B in the wild type ovary, where identification of these structures was difficult with the former microscopy techniques [10]. With current generation confocal microscopy, while carefully observing wild type stage 10B egg chambers, we could clearly detect the presence of a ring canal associated actin array (subsequently referred as RC cables) (Figs 1A and 2), that by location appeared as a distinct population as compared to the other cytoplasmic bundles. As a control we also examined younger egg chambers where presence of the actin baskets was obvious both in wild type and *frl⁵⁹* mutant chambers (S1H–I' Fig). Thus, we propose that the ring canal associated actin baskets, normally forming during the earlier stages of oogenesis, persist continuously till the second phase of dumping, taking place in stage 10B. Moreover, we noticed that while in stage 8 the wild type RC cables are no longer than a few microns (S1H Fig), by late stage 10B they elongate till 20–22 μm, that is sufficient to reach close to the cell nucleus (S2C–C", E Fig). In, the absence of FRL the RC cables are even longer (S2D–D", E Fig) that might represent a mechanism to compensate for loss of the cytoplasmic cables.

To extend the phenotypic characterization of the *frl* mutants, we also investigated older egg chambers, and we detected numerous examples for nuclei blocking the ring canals in stage 11–13 (Fig 1C, C'), revealing a dumping problem that explains well the reduction in egg length. Because we were able to rescue these phenotypes by expressing the full length FRL protein in the germ cells (Fig 1D–F, H), we conclude that FRL is required for cytoplasmic actin cable assembly in the nurse cells. In accordance with this, the FRL protein is strongly accumulated at the nurse cell plasma membranes, as well as at the ring canals (Fig 2). Higher resolution images from stage 10B show that FRL displays a punctate pattern at the end of the cytoplasmic actin filaments at the nurse cell borders, where it shows a strong colocalization with Ena (Fig 2J–I'), and it also exhibits a weaker staining along the actin cables at their initial segment close to the membrane (Figs 1J, K and 2G–I'). Likewise, we detected similar FRL-Ena puncta around the ring canals, located on either sides of the actin rings (with the exception of the oocyte-nurse cell border), and tipping the actin bundles that extend towards the nucleus (Fig 2J–L'). These actin cables do not appear to be directly connected to the ring canal proper (although they initiate from the immediate vicinity) (Fig 1L–L"), instead, distribution of the FRL-Ena dots indicate that they presumably originate from similar microvilli-like protrusions as the cytoplasmic actin filaments (Fig 2). Together, these data suggest that FRL is enriched at the (+) end of the actin filaments (marked by Ena) both in the nurse cell membrane protrusions and at the ring canals. Despite being present in both regions, FRL is only indispensable for assembly of the cytoplasmic actin cables, but not for the ring canal associated ones.

## Formin redundancy has a minor contribution to nurse cell actin cable formation

Because formins, including FRL, were reported to act in a redundant manner in some instances [22–24], we wanted to address whether the ring canal derived actin cables (still present in the absence of FRL) are dependent on another formin. To this end, the other five *Drosophila* formins were knocked down in an *frl⁵⁹* mutant background with a germline specific driver matα4-Gal4-VP16 (*mat-tubG4*). As a control, the formin TRiP lines were first examined in a wild type background, but they had no effect on egg size or actin organization in the nurse cells (S3C Fig), while the knockdown of FRL had a similar, albeit weaker effect both on cytoplasmic actin organization (S3A, B Fig) and egg length (S3C Fig) as that of the

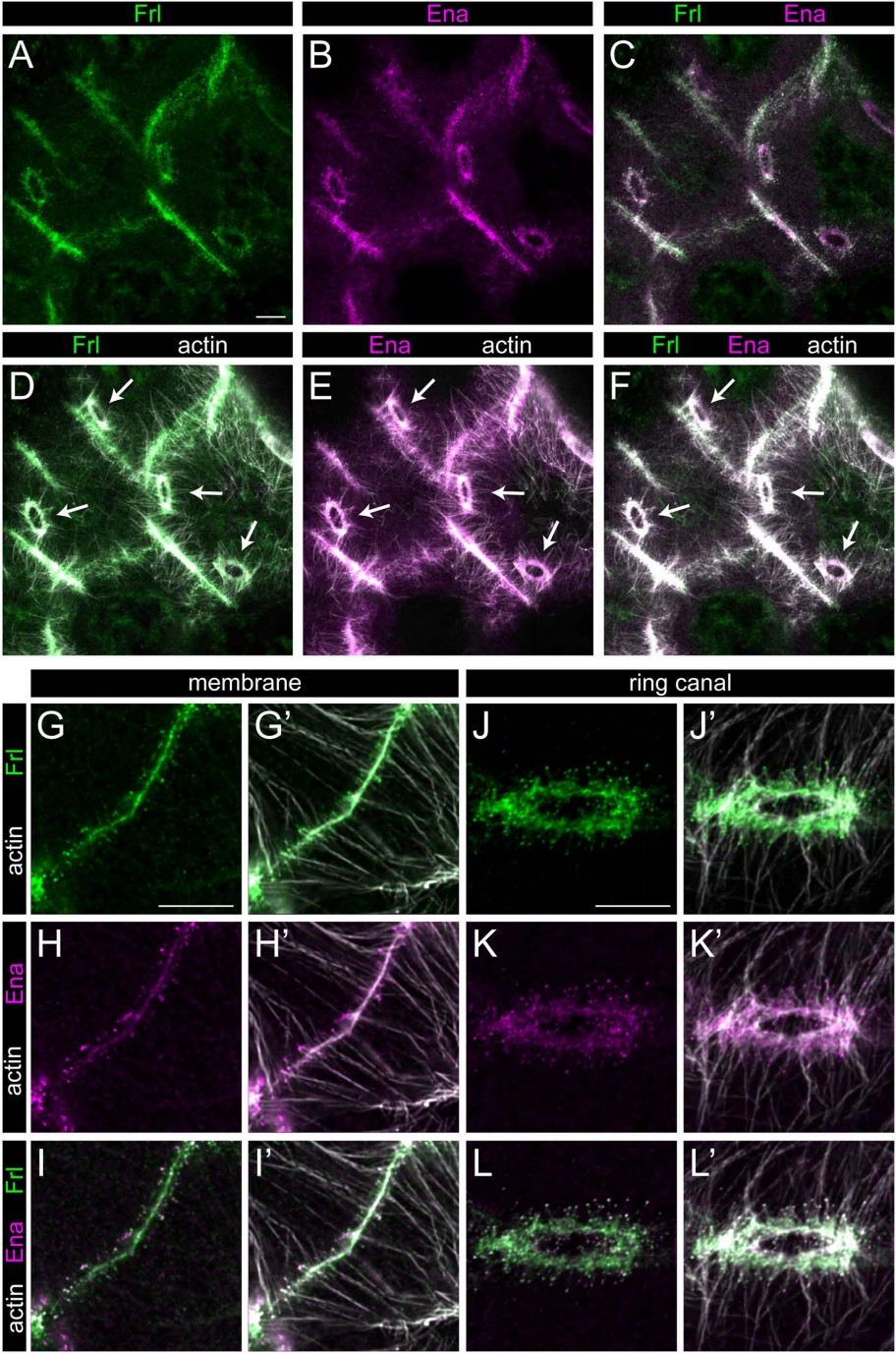

**Fig 2. The localization pattern of FRL and Ena in wild type egg chambers. (A-E)** Confocal Z-sections from a wild type stage 10B egg chamber stained for actin (in gray), FRL (in green) and Ena (in magenta). Panels A-C display a single optical section where colocalization of actin, FRL and Ena is evident at the cortical membrane as well as at the ring canals. D-F show Z-sections of the same area as depicted in A-C, revealing a large number of actin bundles including the ring canal associated cables (arrows) and the plasma membrane attached ones. The strong colocalization of actin, FRL and Ena is also evident. **(G-L')** High magnification images of the plasma membrane (G-I') and ring canal (J-L') accumulation of FRL (in green) and Ena (in magenta). FRL and Ena both show a cortical membrane enrichment (FRL exhibiting a higher level than Ena), and both proteins accumulate in a punctate pattern at the tip of nuclear positioning actin bundles (G-I'). Similarly, FRL and Ena colocalize at the ring canals, and they decorate the tip of the RC cables (on both sides of the actin rings) (J-L'). Scale bars, 10 μm.

*frl⁵⁹* allele. When the knockdown was carried out in the *frl⁵⁹* homozygous mutants, we detected a reduction of egg length in the cases of *capu*, *dia*, *form3* and *DAAM*, while *fhos* had a negligible effect (Fig 3G). However, unlike we expected, none of these mutant combinations exhibited a notable effect on the RC actin cables as compared to the *frl⁵⁹* mutant controls (Fig 3A–F), indicating the lack of redundancy at this level. Nevertheless, comparison of the actin filament patterns in stage 10B uncovered a considerable strengthening of another aspect of the *frl⁵⁹* phenotype evident in the oocyte membrane associated cables. In the mere absence of FRL, it is typical that loss of the cytoplasmic actin cables is nearly complete in the anterior and middle regions of the egg chambers, whereas the effect is weaker along the oocyte membrane where numerous cables can be detected in the posterior most nurse cells that are in direct contact with the oocyte (Fig 3A, H). The concomitant loss of *frl* and *capu*, *dia*, *form3* or *DAAM* decreases the number of these cytoplasmic actin cables without affecting the ring canal derived subpopulation (Fig 3B–E, H), and consistent with the reduced egg size (Fig 3G), number of the ring canals blocked with nucleus is increased in these conditions. Thus, based on the analysis of all six *Drosophila* formins in single and double mutant combinations, we conclude that the ring canal derived nucleus positioning actin cables are unlikely to be formin dependent, whereas formation of the cytoplasmic bundles is primarily dependent on FRL, with a minor contribution of Capu, Dia, Form3 and DAAM, mainly promoting assembly of the oocyte cables.

## Cdc42 and FRL work together in the nurse cells

FRL is a member of the Diaphanous related formin (DRF) family, the activity of which is known to be regulated by Rho GTPases [25]. To determine which *Drosophila* GTPase might be involved in FRL activation during oogenesis, we analyzed four members of the Rho family by RNAi mediated silencing. We found that germline specific knockdown of RhoA and Cdc42 resulted in a significant reduction in egg size, whereas Rac1 and RhoL silencing had no effect (Fig 4G). Next, we inspected the actin cytoskeleton in these ovaries, and noticed that in the knockdown of RhoA the cytoplasmic actin filaments looked largely normal, but the ring canals appeared distorted as indicated by their actin organization and Hts distribution (S4 Fig). Clearly distinct from this effect, the egg chambers upon Cdc42 knockdown looked similar to the ones observed in *frl⁵⁹* homozygous mutants. We detected a strong reduction in the number of the cytoplasmic actin cables (Fig 4H, I), whereas the RC actin bundles were often longer and more prominent than in wild type (Fig 4A–B'). To further confirm the effect of Cdc42 RNAi, we examined the actin phenotypes in mutant clones of *cdc42⁴* (a strong LOF allele) [26]. These studies corroborated the strong effect on the cytoplasmic actin bundles, which were almost entirely missing, and although the RC cables were present, they did not show an increased size or actin level, instead they looked less compact than in controls, and they failed to prevent clogging of the ring canals (Figs 4C, C' and 6C–E). Therefore, a weaker reduction of Cdc42 level (induced by RNAi) produces a nearly identical effect as the complete loss of FRL, whereas the stronger allele not only prevents formation of the cytoplasmic actin cables, but it partly blocks growth of the ring canal derived filaments as well. Given that the *frl⁵⁹* and *cdc42²* mutants exhibit a dominant genetic interaction as judged by cytoplasmic actin cable density and the effect on egg length (Fig 4D–F', J–L), these findings suggest that Cdc42 is required for FRL (and possibly other DRF) activation during cytoplasmic actin assembly, and it also plays a role in RC cable regulation. To address this possibility further, we asked whether Cdc42 KD affects the distribution of FRL and Ena in the nurse cells. These studies showed that the cortical membrane accumulation of FRL and Ena, and the RC accumulation of FRL are strongly reduced upon RNAi mediated silencing of Cdc42 (S5A–O' Fig), while the Ena level appears moderately reduced at the RCs (S5P, P' Fig). Thus, Cdc42 contributes to the regulation of FRL localization in nurse cells, providing additional support for a Cdc42 dependent formin activation in the context of nuclear positioning. Likewise, Ena localization is also affected offering a possible explanation for the seemingly stronger effect of Cdc42 KD as compared to *frl* loss.

## The differential effect of the lack of FRL and Ena on cytoplasmic actin

After establishing a Cdc42 dependent role for FRL in nurse cell actin regulation, we aimed to clarify further the contribution of FRL and Ena. Former studies showed that in the absence of Ena, formation of the nurse cell cytoplasmic actin

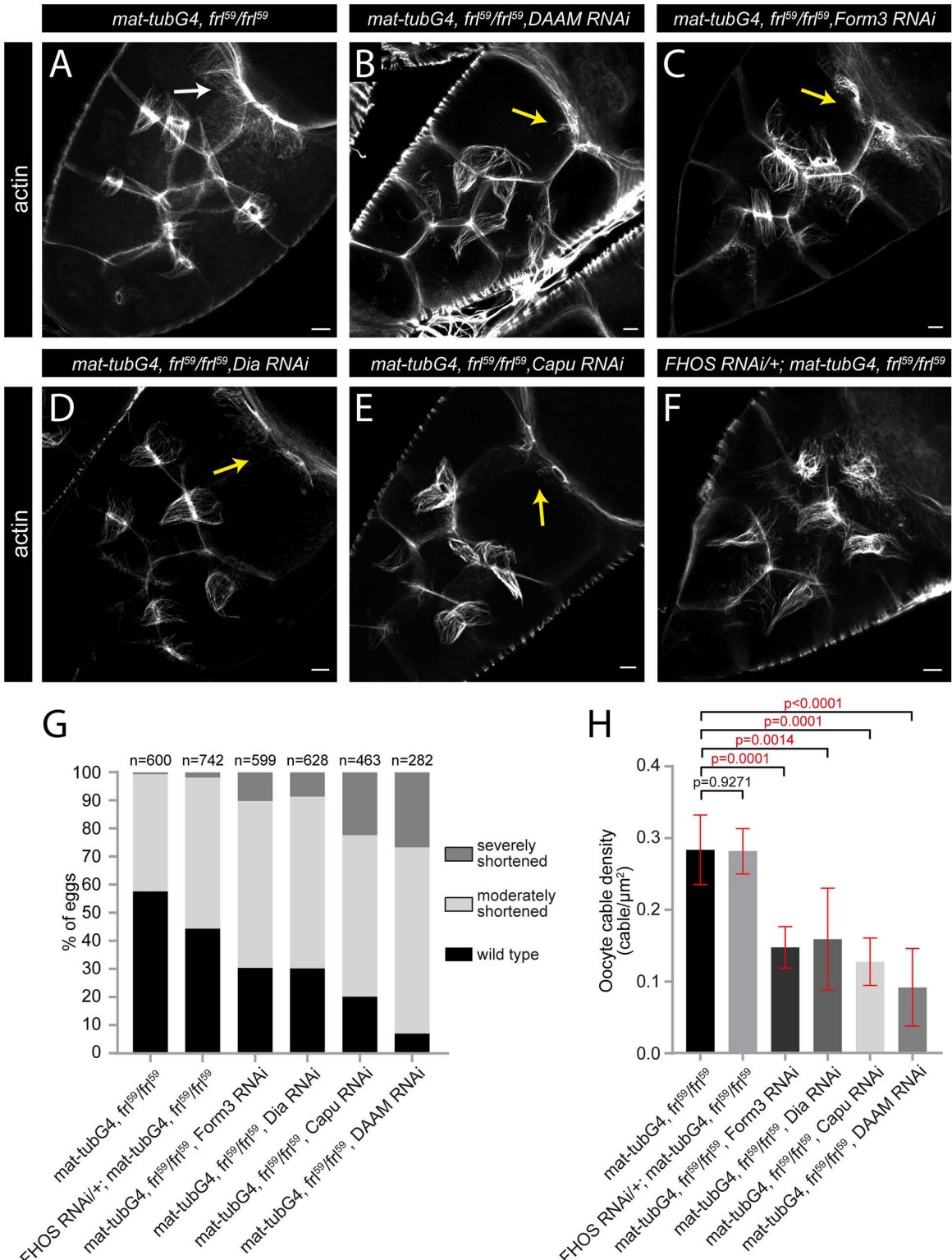

**Fig 3. Formin redundancy in nurse cell actin cable formation. (A-F)** Actin staining of stage 10B egg chambers of the indicated genotypes. In control (*mat-tubG4, frl⁵⁹/frl⁵⁹*) **(A)** egg chambers most of the cytoplasmic actin cables are absent, only a few of them are visible in the oocyte nurse cells (arrow), whereas the RC bundles can clearly be detected. Upon RNAi mediated knockdown of DAAM **(B)**, Form3 **(C)**, Dia (**D**) or Capu (**E**) in an *frl⁵⁹* mutant

background the RC cables remain visible, however, the oocyte cables are often completely missing (yellow arrows in B-E). The knockdown of FHOS has a negligible effect on nurse cell cytoplasmic actin organization **(F)**. **(G)** Quantification of the egg length phenotype laid by mothers with the indicated genotypes. *n* indicates the number of eggs measured. **(H)** Quantification of oocyte cable density in stage 10B egg chambers of the indicated genotypes. Scale bars, 10 μm.

filaments is impaired [19], and we have shown here that these two proteins colocalize in the egg chambers (Figs 1G, 2), inspiring us to investigate whether they act together or they have independent roles in the process. To this end, we decided to compare the effects of the loss of FRL and Ena by using *frl⁵⁹* for FRL and the FP4mito tool to deplete Ena in the female germline. Expression of FP4mito was shown to rapidly relocalize Ena to mitochondria [27], and it has already been successfully used to mimic the loss of function phenotype of *ena* within the *Drosophila* ovary [19]. Accordingly, our immunostaining experiments confirmed that upon FP4mito expression with *mat-tubG4*, Ena is absent from the nurse cell plasma membranes and the ring canals (Fig 5A, B, E, G, H, K). As a consequence of the lack of Ena, we found that the eggs laid by *UAS-FP4mito/ +; mat-tubG4/ +* mothers are shorter than the controls and even that of the *frl⁵⁹* mutant eggs (Fig 5R, compare to Fig 1F). Whereas at this point one could assume that Ena might have a stronger effect on the cytoplasmic actin network than FRL, comparison of the *UAS-FP4mito/ +; mat-tubG4/+* and *frl⁵⁹* mutant egg chambers led us to a different conclusion. We revealed that in the absence of Ena the cytoplasmic actin cables are relatively mildly affected (quantified in 5S, T Fig), while the ring canal linked actin cables are largely missing or appear much shorter than in the wild type situation (Figs 5M, N, 6B, B'). It is noteworthy, that the actin bundles around the posterior ring canals (the ones closer to the oocyte) are more strongly affected than the anterior ring canals (Fig 6A, C–E). Moreover, we also noted that blocking of the ring canals by nuclei occurs more frequently upon Ena depletion (Fig 6B') (typically 4–5 blocking events per egg chamber are detected already at stage early 10B) than in the *frl* null mutants (with 1–2 blocking events, only visible in late stage 10B). Importantly, when clogging occurs in the absence of Ena, the cytoplasmic cables can still be detected in the nurse cells (Fig 5O–Q'), revealing that the presence of these actin bundles is not sufficient for normal nuclear positioning. Additionally, the effect of Ena depletion was analyzed in younger egg chambers as well, where we failed to detect the presence of the ring canal actin baskets (S1J, J' Fig).

To collect an independent line of evidence for the effect of Ena, a germline clone analysis was carried out with *ena²³*, a hypomorphic allele reported to cause dumping defects [19]. We observed that this mutation mildly impairs the cytoplasmic actin cables, and we also detected an occasional loss or strong reduction of the RC associated cables (S6 Fig). Penetrance of the RC cable phenotype is clearly weaker than upon the FP4-mito mediated depletion of Ena, and accordingly, the dumping defect is also weaker (S6J Fig) [19]. However, unlike in the case of Ena depletion where Ena level is severely reduced (causing a nearly protein null situation in the cytoplasm), *ena²³* is not a null allele because the Ena protein can still be detected in the germline clones with a distribution pattern that is similar to wild type (shown in S6D–I Fig). Consistent with this, *ena²³* is associated with two mutations, N379F and K636STOP [28], resulting in a truncated protein lacking the tetramerization domain but retaining the actin binding region. Thus, the hypomorphic nature of *ena²³* and the largely normal protein accumulation pattern provide a conceivable explanation for the phenotypic difference as compared to Ena depletion, and overall, the data support further for an Ena role in RC associated actin bundle formation.

Collectively, these data suggest that the depletion of Ena results in an actin phenotype which is largely complementary to the lack of FRL, because Ena has a much stronger effect on the RC cables with a modest effect on the cytoplasmic filaments, while *frl* strongly affects this latter actin subpopulation without preventing the formation of the ring canal attached cables (compare Figs 1B, D and 5M). Thus, phenotypic characterization of these two functionally important actin regulatory factors revealed that, beyond the clear difference in the position of the cytoplasmic and the RC actin cables, these two nuclear positioning actin subpopulations differ in their assembly mechanisms as well, one of them being primarily Ena dependent while the other one being FRL dependent. Interestingly, although the lack of the FRL dependent cables seems to induce a more obvious reduction in the cytoplasmic actin level, the lack of Ena results in stronger dumping defects (i.e.,

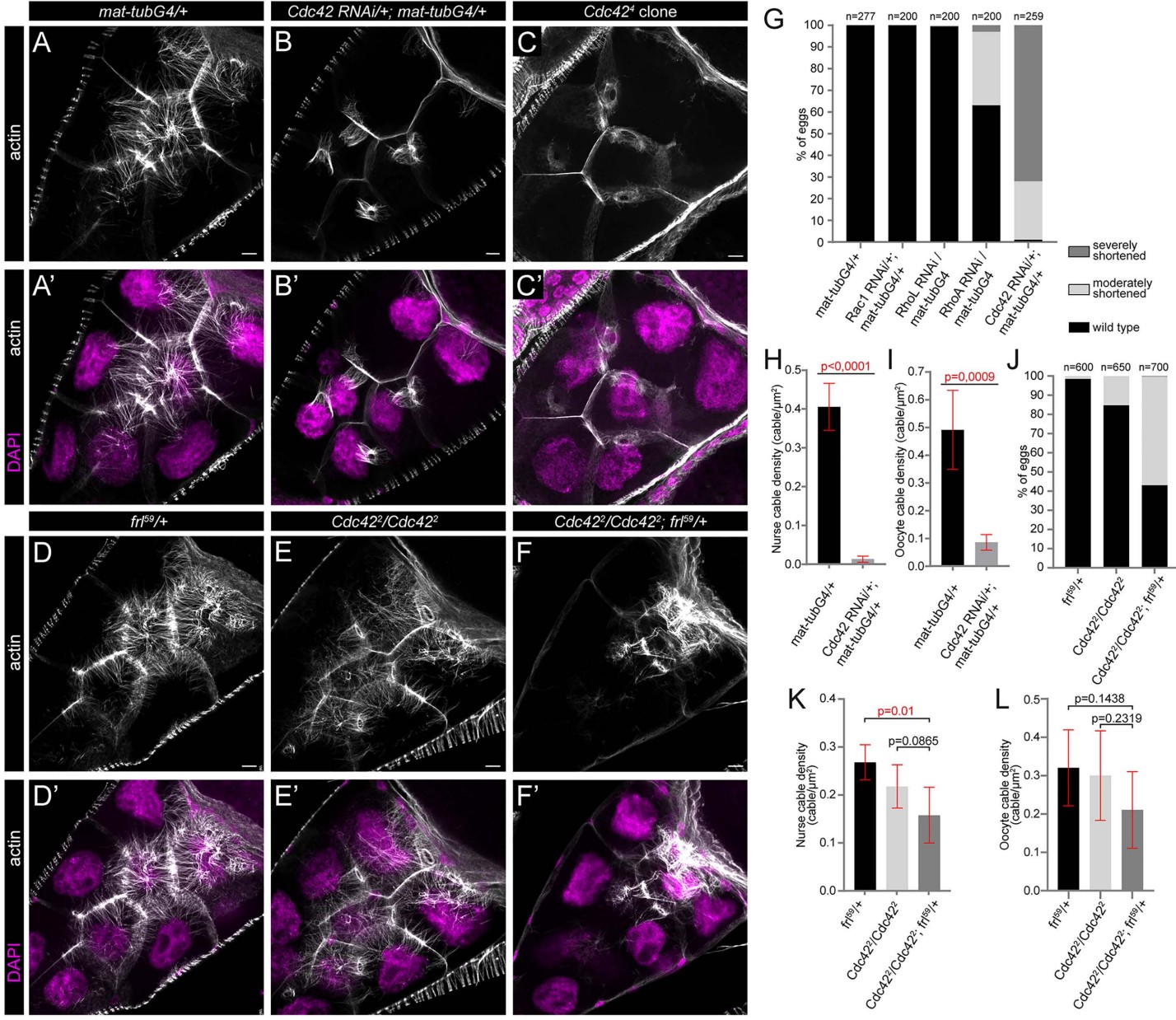

**Fig 4. The loss of Cdc42 impairs nuclear positioning actin in a similar manner as _frl_.** (**A-C'**) Confocal Z-sections of stage 10B egg chambers stained for actin (in gray) and DAPI (to label the nuclei, in magenta). As compared to control (_mat-tubG4/+_) (**A, A'**) egg chambers, knockdown of Cdc42 results in a complete absence of the cytoplasmic actin bundles, whereas the RC cables often look longer and more prominent than in wild type (**B, B'**). In _Cdc42⁴_ mutant germline clones the cytoplasmic actin filaments are entirely missing, while the ring canal associated actin cables look less compact than in controls and often fail to prevent clogging of the ring canals (**C, C'**). (**D-F'**) Heterozygosity for _frl⁵⁹_ enhances the weak cytoplasmic cable phenotype of _Cdc42²_ mutant egg chambers. Note the reduced cytoplasmic actin level in **F. (G, J)** Quantification of the egg length phenotype laid by mothers with the indicated genotypes. _n_ indicates the number of eggs measured. (**H, I, K, L**) Quantification of nurse cable (H, K) and oocyte cable (I, L) density in egg chambers with the indicated genotypes. Note the strong reduction of cable density upon Cdc42 KD in both cytoplasmic cable populations, and enhancement of the phenotypic effect of _Cdc42²_ by _frl⁵⁹/+_. Scale bars, 10 μm.

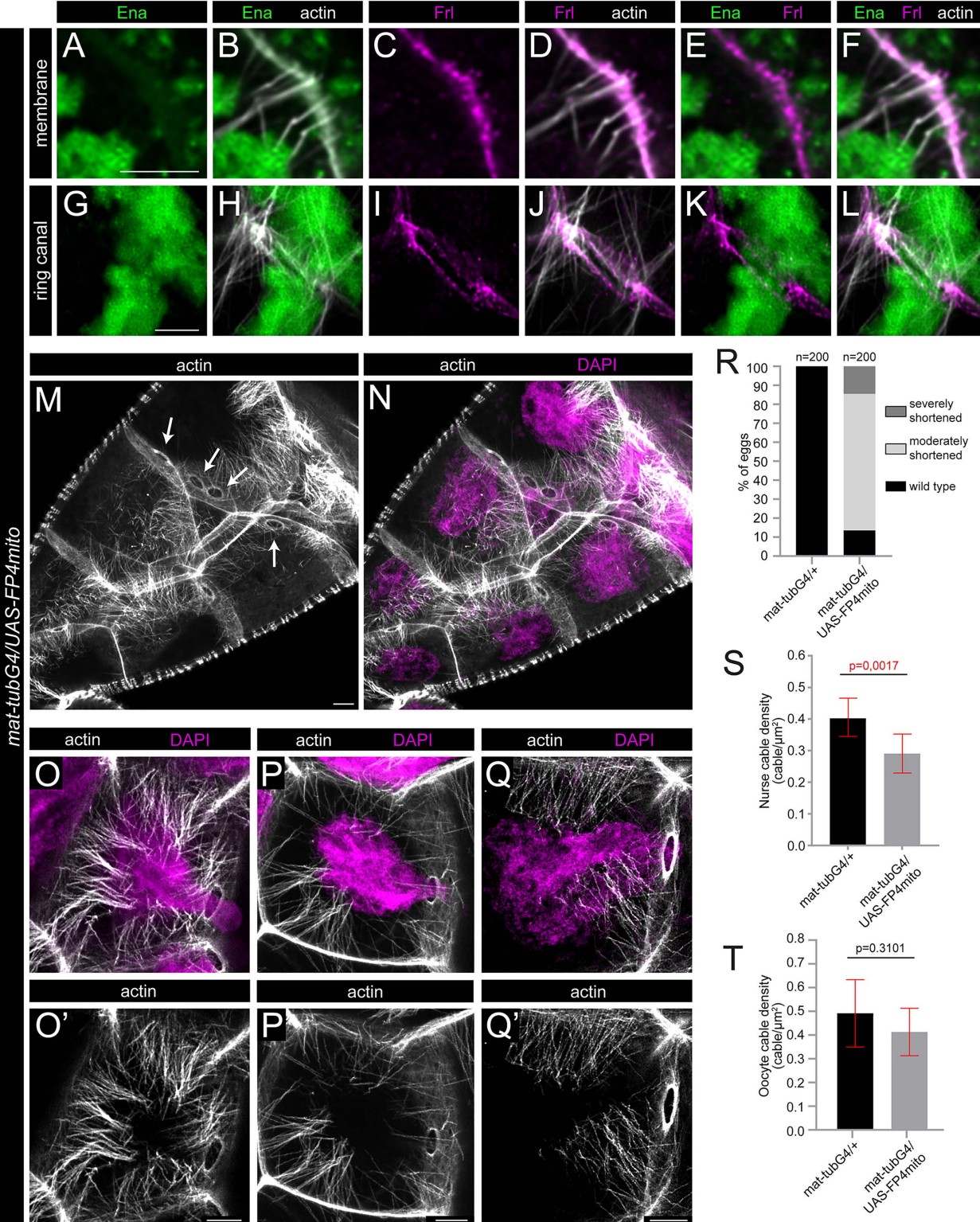

**Fig 5. The depletion of Ena strongly reduces number of the ring canal associated actin cables. (A-L)** Confocal Z-sections of a nurse cell plasma membrane area (A-F) or a ring canal (G-L) in a *mat-tubG4/UAS-FP4mito* stage 10B egg chamber, stained for actin (in gray), Ena (in green) and FRL (in magenta). Note the lack of Ena along the membrane **(A, B)**, and the presence of FRL both at the tip of the actin filaments and along the membrane

**(C-F)**. In *mat-tubG4*/*UAS-FP4mito* egg chambers the RC cables are often lost or reduced in number, and Ena is absent from the ring canals **(G, H)**, while FRL is present in a reduced amount **(I-L)**. **(M, N)** Actin (in gray) and nuclear (DAPI, in magenta) staining of a *mat-tubG4*/*UAS-FP4mito* egg chamber at stage 10B, revealing the presence of a high number of cytoplasmic nuclear positioning actin cables, paralleled with the frequent loss of the RC cables (arrows in **M**). **(O-Q')** Three different examples to illustrate that despite presence of a large number of cytoplasmic cables, the nuclei (in magenta) often clog the ring canals (missing the RC cables) in *mat-tubG4*/*UAS-FP4mito* egg chambers. **(R)** Quantification of the egg length phenotype laid by mothers of the indicated genotypes. *n* indicates the number of eggs measured. **(S, T)** Quantification of nurse cable and oocyte cable density in *mat-tubG4*/*UAS-FP4mito* egg chambers, revealing a moderate reduction in both cases. Scale bars: 10 µm in panel M-Q', 5 µm in all other panels.

shorter eggs are formed). Based on these findings, we propose that the RC cables, present in the *frl* mutants but largely absent from the Ena depleted egg chambers, play a more prominent role in keeping the nuclei away from the ring canals than the cytoplasmic actin bundles.

## The concurrent absence of FRL and Ena prevents cytoplasmic actin formation completely

To extend our studies on the regulatory connection and differential contribution of FRL and Ena, we looked into the localization of these proteins in wild type egg chambers and also in the absence of one another. In the wild type situation both proteins are present from the early stages of oogenesis, but their nurse cell plasma membrane association is relatively weak (shown for stage 4 in S7A–B" Fig). By stage 8 FRL and Ena clearly show a cortical membrane association (S7C–D" Fig), and Ena also exhibit a strong accumulation at the ring canals with a punctual pattern, where the FRL level remains modest (S7E–F" Fig). By contrast, in stage 10B colocalization of the two proteins becomes more prominent both along the plasma membrane and at the ring canals (S7G–L" Fig). Hence, FRL and Ena display a significant level of colocalization at the nurse cell cortex during the entire course of oogenesis, but FRL is largely absent from the ring canals up to stage 8, which is in good accordance with their differential requirement in RC cable formation.

Next, we asked whether the lack of *frl* affects the accumulation and localization of Ena in the developing egg chambers. The enrichment of Ena at the nurse cell plasma membranes is greatly reduced in the *frl*[59] mutants, whereas the Ena signal remains strong at the ring canals (Fig 7A–D'), including the tips of the RC bundles (S8 Fig). This observation suggested that the effect of *frl* might be related to the reduced Ena level at the plasma membranes. If so, one would expect that increasing the level of Ena in an *frl* mutant background would rescue (or partly rescue) the phenotype. However, the overexpression of Ena in *frl*[59] mutants had no such an effect (Fig 7I), although the Ena::Flag protein expressed has the ability for membrane association (S9 Fig). To estimate the reduction of Ena level in *frl*[59], an *ena* RNAi line was used for comparison, and we detected a similar decrease in both cases (Fig 7G, H). Despite the comparable protein levels, the knockdown of Ena has very weak if any effect on egg size or cytoplasmic actin organization (Fig 7E–F', J, K), indicating that the primary effect of *frl* is not directly linked to a drop of Ena levels. Besides analysis of the *frl*[59] mutants, we also considered the reverse setting when we determined the FRL protein distribution upon Ena depletion by FP4mito, however, we found no major change in the amount or localization pattern of FRL in this condition (as compared to wild type) (S10 Fig). Taken together, these data revealed that FRL is required for the plasma membrane/actin (+) end enrichment of Ena but not for its ring canal association, whereas FRL distribution in the nurse cells does not appear to be regulated by Ena. Moreover, these results provided further support for a specific FRL requirement in cytoplasmic actin assembly, which is clearly distinct from the role of Ena.

The existence of an FRL and an Ena dependent subclass within the nucleus positioning actin cables encouraged us to ask whether these two arrays are the only ones required, and to verify further the complementary role of these two proteins. To address these issues, we first carried out a genetic interaction assay between *frl* and *ena* by generating *ena*[23]/+; *frl*[59] females. These mothers laid significantly shorter eggs compared to homozygous *frl*[59] controls (Fig 8E), further confirming the essential role of these two genes. Next, we depleted Ena in a homozygous *frl*[59] mutant background, and we not only observed a strong genetic interaction as to egg length reduction (Fig 8E), but the concomitant absence of FRL and Ena resulted in a complete absence of both the cytoplasmic and the RC actin cables (Fig 8A, A'). Despite of this

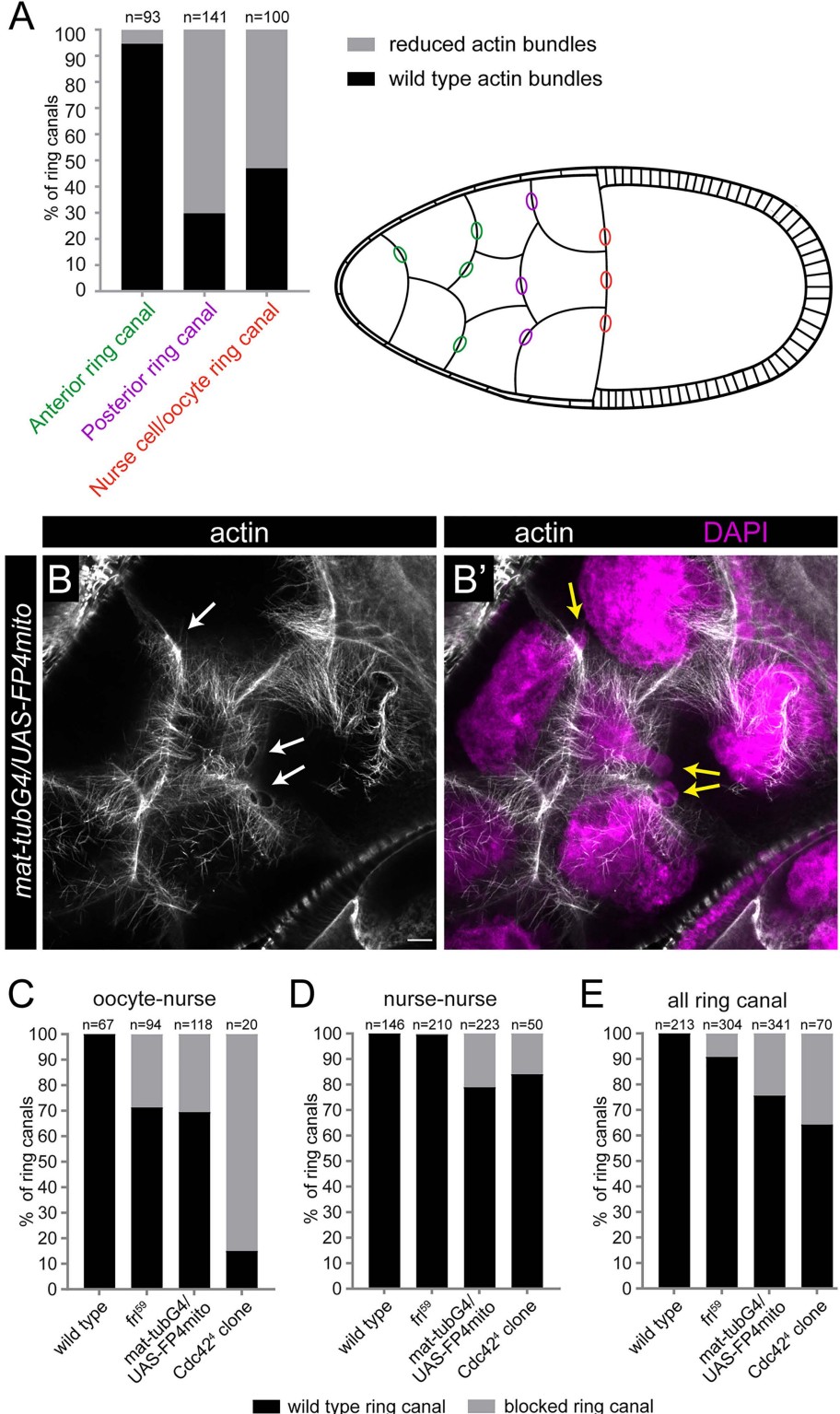

**Fig 6. Ena is primarily required for formation of the RC cables. (A)** Quantification of the ring canal actin defects observed upon Ena depletion with FP4mito. Note that the lack of Ena results in reduced actin levels at the ring canals, particularly in the cases of the posterior and oocyte ring canals, whereas the anterior ring canal cables are much less affected. **(B, B')** Actin (in gray) and nuclear (DAPI, in magenta) staining of a

*mat-tubG4*/*UAS-FP4mito* egg chamber at stage 10B, revealing the presence of a high number of cytoplasmic cables, paralleled with the frequent loss of the ring canal associated bundles (arrows in B) causing ring canal blocking events (yellow arrows in B'). **(C-E)** Quantification of the ring canal blocking events in stage 10B egg chambers with the indicated genotypes. Scale bars, 10 µm.

remarkable effect, the overall shape of the nurse cells and that of the ring canals remained largely normal (Fig 8A'), even though dumping was almost entirely blocked, as indicated by the large number of clogging events (Fig 8A–D) and the small egg size (hardly bigger than a stage 11 oocyte) (Fig 8E). Thus, analysis of the egg chambers with strongly reduced Ena and FRL levels provided strong support for non-overlapping FRL and Ena roles in nucleus positioning cytoskeleton assembly. Strikingly, the simultaneous loss of these two proteins entirely prevented the formation of both nuclear positioning actin subpopulations (i.e., the RC cables and the cytoplasmic actin cables), not only arguing that FRL and Ena are the main regulators of nucleus positioning actin, but also suggesting that functionally these are the only two main components.

## Discussion

The *Drosophila* egg chambers have long been used as a paradigm to study the mechanisms of nuclear positioning, including that of the oocyte nucleus, the nurse cell nuclei and the follicular cell nuclei [5,29–31]. Of these, we focused on the nurse cell nuclei that need to be kept away from the ring canals during the last stages of oogenesis to allow the bulk transport of materials towards the oocyte. According to the prevailing model this is achieved by a dense network of cytoplasmic actin cables, assembled shortly before dumping, that anchors nurse cell nuclei in the central cytoplasm. Our findings refine this model by demonstrating that nuclear positioning is mediated not by a single actin array but by two distinct filament populations which differ in their position (cytoplasmic *versus* ring canal attached), time of origin (early *versus* late stage of oogenesis), as well as in the molecular machineries that build them.

The nucleus positioning cytoplasmic actin filaments form during stage 10B in microvilli-like membrane pits, and grow inward as bundles of ~25 filaments to extend until the nuclear area, while reaching a considerable length of about 30 µm. We demonstrate that their assembly is highly dependent on the formin FRL, likely acting downstream of Cdc42. FRL loss results in a clear reduction of the cytoplasmic cables, and the phenotype is only minimally enhanced by depletion of other formins, indicating that FRL is the principal assembly factor for this actin population. The ring canal associated actin baskets were initially described to form during stage 6–7 and persist until stage 10 [11], however, their identification during the later stages was made difficult by appearance of the cytoplasmic cables [10], and they have not been linked to nuclear positioning. By using high resolution imaging, we could demonstrate their presence even at stage 10B in wild type egg chambers, which is supported further by their undoubted appearance upon the largely selective destruction of the cytoplasmic cables via the *frl* null mutation. These findings suggest that the RC actin baskets contribute to the regulation of dumping in at least two ways, first by controlling directional sorting of vesicles and potentially other cargos during the slow cytoplasmic streaming phase as proposed by Nicolas et al., 2009 [10], and also during stage 10B when they participate in the prevention of nuclear clogging. Thus, it appears that nucleus positioning during the rapid late phase of dumping is not simply achieved by the newly forming cytoplasmic actin arrays, but rather by the combined action of the ring canal actin baskets (formed well before stage 10B) and the cytoplasmic cables. This conclusion is fully supported by our mutant analysis when one or the other system is selectively impaired. Nevertheless, the question arises as to whether the RC cables indeed play a role in the wild type situation as well. Given that upon Ena depletion most of the cytoplasmic cables are present, yet the nuclei often block the ring canals, this situation clearly indicates that presence of the cytoplasmic cables is not sufficient to hold the nuclei away from the RCs. By taking this together with the position and length of the RC cables, long enough to span the distance between the ring canals and the nucleus in wild type nurse cells, our observations strongly support for the RC cables being required for nucleus positioning not only in the *frl* mutants, but also

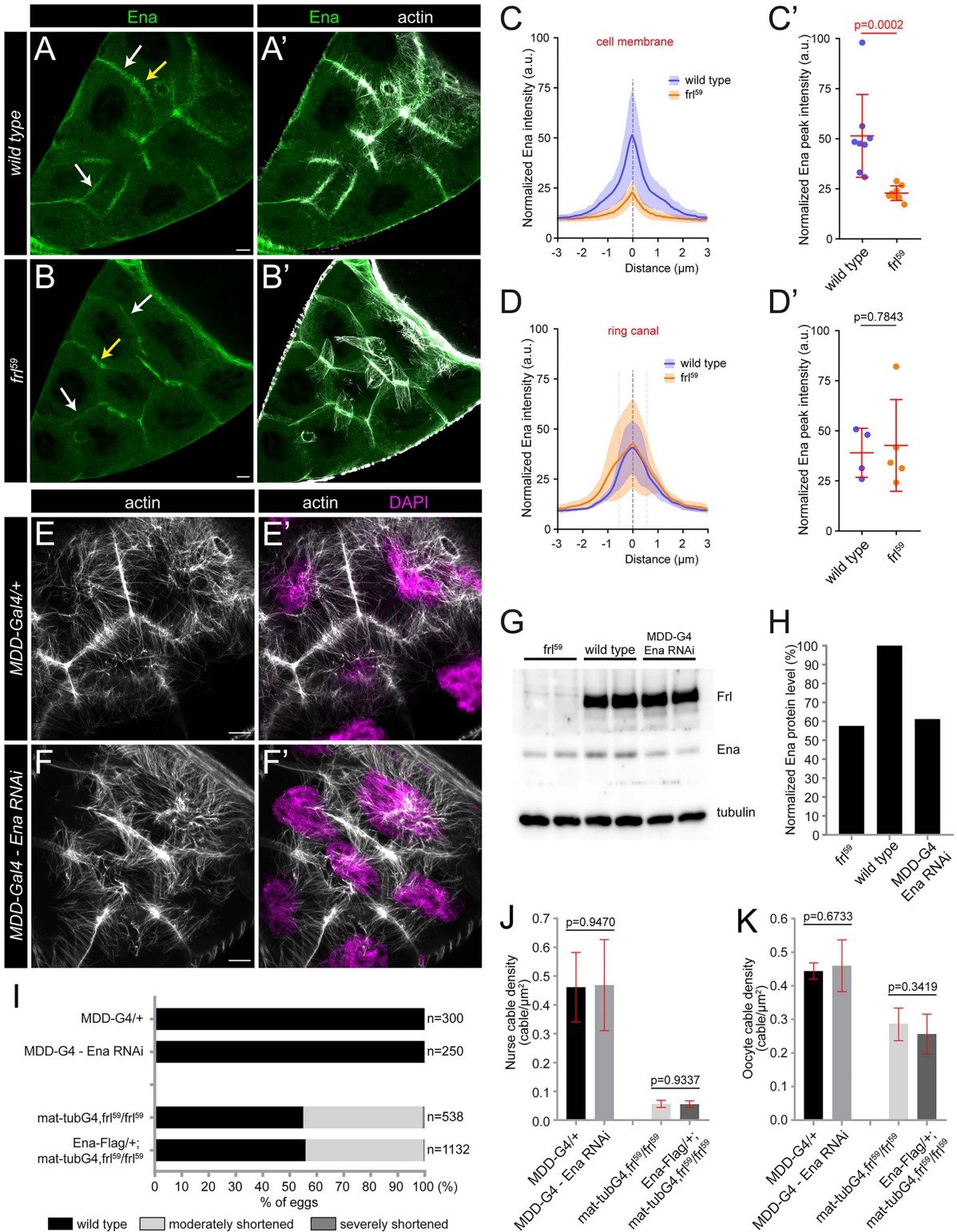

**Fig 7. FRL is required for nurse cell plasma membrane accumulation of Ena. (A-B')** Confocal Z-sections of a wild type (A, A') and an *frl⁵⁹* mutant (B, B') egg chamber stained for Ena (in green) and actin (in gray). Note that in the *frl⁵⁹* mutant Ena exhibits a strongly reduced level along the plasma membrane (white arrows indicate two typical examples), while its ring canal enrichment is not impaired (yellow arrows point to ring canal examples).

(**C-D'**) Quantification of Ena levels at the plasma membrane (C, C') and at the ring canals (D, D') in wild type and *frl⁵⁹* mutant egg chambers underlines the conclusion shown in A-B'. (**E-F'**) The RNAi mediated knockdown of Ena has a negligible effect on nuclear positioning actin in stage 10B egg chambers. (**G**) Western blot analysis of the protein expression level of FRL and Ena in wild type, *frl⁵⁹* and *MDD-G4/Ena-RNAi* egg chambers. Note the similarly reduced Ena levels in *frl⁵⁹* and *MDD-G4/Ena-RNAi*, which is quantified in **H**. (**I**) Quantification of the egg length phenotype laid by mothers of the indicated genotypes. *n* indicates the number of eggs measured. (**J, K**) Quantification of nurse cable (J) and oocyte cable (K) density in egg chambers with the indicated genotypes. Note that expression of Ena-Flag fails to rescue the defects caused by the *frl* mutation. Scale bars, 10 μm.

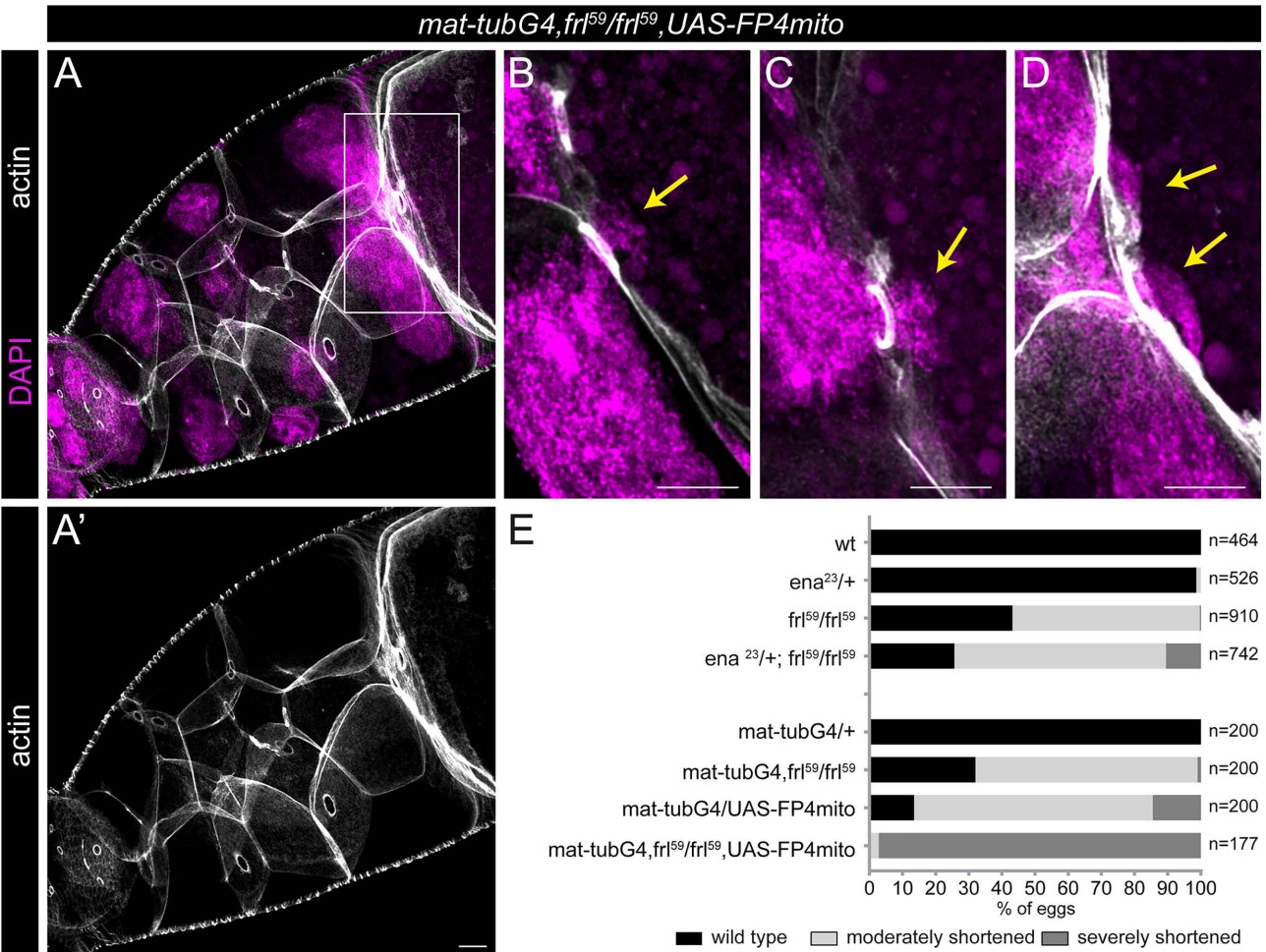

**Fig 8. Egg chambers forming upon the concomitant absence of FRL and Ena are completely devoid of nucleus positioning actin cables. (A-D)** Confocal Z-sections of a *mat-tubG4, frl⁵⁹/frl⁵⁹, UAS-FP4mito* stage 10B egg chamber stained for actin (in gray) and DAPI (to label the nuclei, in magenta). Note the presence of the nurse cell cortical actin and the actin filaments at the inner rim of the ring canals (A'), and the complete absence of the nuclear positioning actin bundles (including the cytoplasmic and the ring canal associated subpopulations). The inset in B-D highlights nuclear clogging events (yellow arrows), frequently occurring upon the concomitant absence of FRL and Ena. (**E**) Quantification of the egg length phenotype laid by mothers of the indicated genotypes. *n* indicates the number of eggs measured. Scale bars, 10 μm.

in wild type conditions. In addition, because the lack of Ena results in a stronger dumping defect than the lack of FRL, our data argue for a more prominent role for the RC cables in the prevention of clogging than that of the FRL dependent cytoplasmic cables. Thus, although the nuclear positioning actin cables execute a single major function, and thereby they can be viewed as a uniform system, this complex array can in fact be subdivided into two subpopulations, that differ in

the temporal and spatial regulation of their initiation, and their contribution to dumping efficiency. Whereas we noticed that Ena depletion has a weak effect on the cytoplasmic cables too, the differential and region-specific requirement for FRL and Ena is still evident in the phenotypes, primarily supporting for non-overlapping functions. When Ena is depleted in an *frl* null mutant background, both actin populations are eliminated without affecting cortical actin or nurse cell architecture. This is an unusually strong and clean phenotypic effect that has not been reported in former studies, paralleled with frequent clogging of the ring canals and a very strong defect in dumping. The eggs laid by these mothers are barely longer than the average length of the oocyte within stage 11 egg chambers, indicating that dumping is indeed almost completely blocked. These results imply that the FRL- and Ena-dependent pathways together constitute the entire actin assembly machinery dedicated to nuclear positioning.

The known biochemical activities of these regulators align well with our *in vivo* observations. Formins, including FRL/ FMNL, act as actin nucleators and processive elongation factors, and some also bundle filaments [32]. Ena/VASP proteins promote barbed-end elongation and can facilitate filament bundling [33,34]. Cdc42 is a broad regulator of actin-based membrane protrusions and can activate both formins and Ena/VASP family proteins [32–34]. *In vitro*, FMNL2/3 behave as relatively weak nucleators but can processively elongate actin filaments in the presence of Profilin; and *in vivo* these proteins robustly promote filopodia assembly [35–39]. Several studies place FMNL proteins at the extreme tips of filopodia, where they associate directly with the plasma membrane [40,41], and implicate them in generating convex membrane curvature together with Cdc42 and/or I-BAR proteins [42,43]. Our data indicate that FRL and Cdc42 act as mutually dependent cofactors for assembly of the late, cytoplasmic cables. Although FRL and Ena colocalize at the tips of cytoplasmic cables by stage 10B, the pronounced loss of these cables in *frl* mutants demonstrates that Ena cannot substitute for FRL dependent assembly. Conversely, Ena is required for formation of the RC actin baskets arising at stage 6–7 when Ena exhibits a much stronger accumulation at the RCs than FRL. Ena possesses properties well suited for assembling long, straight, bundled filaments: *in vitro* it can bind and protect multiple new filament barbed ends from capping protein, permitting the establishment of several parallel filaments. These Ena-associated filaments are then efficiently bundled by Fascin, and they can cooperatively extend and maintain a robust filopodia of uniform thickness with aligned barbed ends [44]. It is therefore plausible that a similar Ena - Fascin synergy underlies the assembly of the ring canal actin baskets, consistent with previous identification of Fascin as an important factor [13,45]. Because Ena does not nucleate actin under physiological conditions [33,46,47], additional nucleation factor(s) must initiate filaments of the RC cables. The identity of this factor remains unknown as knockdown of other formins did not produce a ring canal specific defect, nor were such effects reported for *spire* mutants, affecting another nucleation factor family involved in unbranched actin filament formation [48]. These observations leave open the possibility that multiple formins act redundantly at ring canals, or that an as-yet-unidentified nucleator mediates assembly of the RC actin cables. Either way it is, the employment of different nucleation factors for the RC and the cytoplasmic cables would be in accordance with the different temporal and spatial requirements.

Slender, finger-like cellular protrusions with a centrally located actin bundle, such as filopodia, microvilli and the membrane protrusions supporting the nucleus positioning actin bundles in the nurse cells, are widespread in nature. Most filopodia are adhesive, often long and very dynamic (with a half life of not more than few minutes); microvilli in turn are apical, non-adhesive, short and stable extensions, whereas the nurse cell membrane pits exhibit a limited and largely uniform length, and a stability for few hours. Curiously, regardless of these obvious structural differences (accompanied by numerous functional differences, not discussed here), previous studies have already shown that a big part of the molecular toolbox used for the generation of these markedly different structures is similar in all these cases [49–52]. The identification of FRL and Cdc42 as new factors of nucleus positioning in *Drosophila* nurse cells provide further support for this conclusion. This is particularly evident in comparison to filopodia and filopodia-like structures, the formation of which was shown to depend on the FRL orthologs, FMNL2 and FMNL3, in several cellular models [37,42,53–56], and Cdc42 was also identified as a key protein of actin-based protrusion formation in numerous model systems, including FMNL2, FMNL3

and Mena (an Ena/VASP family member) dependent manners [35,43,55,57]. Because ample of evidence confirm the role of Ena/VASP proteins in filopodia assembly in diverse *Drosophila* [58–60] and vertebrate models [55,61–64], we conclude that the three major nuclear positioning factors studied here, are recurrently used elements during evolution to create long actin bundles to provide support for various types of slender cellular protrusions. Yet, beyond these similarities, it remains a largely open question what determines the specificity of these factors in the different cell types and developmental stages, resulting in the formation of seemingly different protrusions. Presumably, the differences in the biochemical properties, the regulation of subcellular localization, and protein-protein interactions are all crucial to adapt a common actin assembly toolkit to distinct structural outputs.

Together our observations provide novel insights into the formation and regulation of the diverse actin arrays dedicated to ensure nuclear positioning in the nurse cells. Whereas identification of a new player and recognition of a two-tier mechanism, are important steps forward, further studies will be required to elucidate the molecular mechanisms of FRL in more details, and to explore how the activities of FRL, Ena and other players of the nuclear positioning system are coordinated during oogenesis to enable the orchestration of the cytoplasmic actin network critical for oocyte development. In addition, because accumulating evidence suggest the presence of actin fibers in mouse cysts before and during cytoplasmic transfer [65], it would be exciting to examine whether analogous mechanisms operate in vertebrate systems.

## Materials and methods

### Drosophila stocks

*Drosophila melanogaster* stocks were raised on standard cornmeal-yeast-agar medium at 25°C. The following mutant strains were used: *w^1118^* (BL#3605), *Cdc42^4^ FRT19A/FM6* (BL#9106), *Cdc42^2^ FRT19A* (BL#9105), *FRTG13 ovo^D1^/T(1;2) OR64/CyO* (BL#4434), *ovo^D1^, hsFLP^12^, FRT19A/C(1)DX* (BL#23880), *hsFlp^12^,CyO/Sco* (BL#1929), *FRTG13 ena^23^/CyO* (BL#25405), *ena^23^/CyO* (BL#8571), *ena RNAi* (BL#39034), *FRL RNAi* (BL#32447), *FHOS RNAi* (BL#51391), *DAAM RNAi* (BL#39058), *Form3 RNAi* (BL#32398), *Capu RNAi* (BL#32922), *Dia RNAi* (BL#33424), *RhoA RNAi* (BL# 32383), *RhoL RNAi* (BL#33723), *Rac1 RNAi* (BL# 34910), *Cdc42 RNAi* (BL#37477), *UAS- GFP-FP4mito* (BL#25747), *mat-tub-Gal4 (P{w[+mC]=matalpha4-GAL-VP16}V37)* (BL#7063), *MDD-Gal4 (P{matalpha4-GAL-VP16}67; P{matalpha4-GAL-VP16}15)* (BL#80361) all from the Bloomington Drosophila Stock Center; *UAS-FRL* [24]; and *frl^59^* [21]. The formin RNAi lines used in this study were already shown to induce effective gene silencing in other tissues [20,24,66–68], hence we consider them as appropriate tools for ovary studies as well.

### Drosophila genetics

The *mat-tub Gal4, frl^59^*; *frl^59^, DAAM RNAi*; *frl^59^, Dia RNAi*; *frl^59^, Capu RNAi* and *frl^59^, Form3 RNAi* lines were generated by standard genetic recombination techniques. The UAS-Ena-Flag construct, containing the short, PA isoform of Ena, was generated using standard molecular cloning techniques, with the pPWF-attB vector and the attP40 landing site. The primers used were "ena forward": 5′-ATGTCGACAATGACTGAGCAGAGTATTATCG-3′ and "ena reverse": 5′- ATGATATCCGTATCTGCGATTAAACTCCG-3′.

### Immunohistochemistry

The females were collected 0–8 hours after hatching and fed with yeast to promote egg production. Their ovaries were dissected two days later in ice-cold PBS and fixed in 4% paraformaldehyde (diluted in PBS) at room temperature for 20 minutes. After fixation, the samples were washed three times in PBS containing 0.1% Triton X-100 (PBST) for 20 minutes each and blocked in PBST with 1% BSA and 5% FCS for 2 hours. Primary and secondary antibodies were diluted in PBST with 1% BSA and 5% FCS and incubated overnight at 4°C. The samples were mounted using ProLong Gold reagent (P36930, Thermo Fisher Scientific). To visualize ring canals in stage 8 and 9, dissection and fixation were performed using cytoskeleton buffer (10 mM MES, pH 6.1, 150 mM NaCl, 5 mM EGTA, 5 mM glucose, 5 mM MgCl$_2$), with fixation carried

out in 4% paraformaldehyde prepared in cytoskeleton buffer. The primary antibodies used in this study were: rat anti-FRL (1:200) [22], mouse α-Ena (1:100; DSHB, 5G2), and mouse anti-hts (DSHB, hts RC). As secondary antibodies, we used the appropriate Alexa Fluor 488- or Alexa Fluor 647-coupled antibodies (1:600; anti-mouse Alexa 488, A-11001; anti-rat Alexa 647, A-21247; Thermo Fisher Scientific). DNA was visualized with DAPI (1:500, Sigma) and actin was labeled with Alexa Fluor 546 (1:50; Phalloidin-Alexa 546, A22283; Thermo Fisher Scientific).

**SiR actin staining**

The females were collected 0–8 hours after hatching and fed with yeast to stimulate egg production. Their ovaries were dissected two days later in ice-cold PBS. For staining, stage 10B egg chambers were isolated and incubated for 1 hour with SiR-actin (1:1000, Spirochrome, CY-SC001, Stein am Rhein, Switzerland) in Schneider's medium supplemented with 10% fetal bovine serum (FBS). Microscopic analysis was performed in glass-bottom Petri dishes (Cell E&G, GBD00001–200).

**Image analysis and quantification**

For embryo length quantification, approximately 50 females of the appropriate genotype were allowed to lay eggs over several days; eggs were collected from three independent egg-laying periods and imaged on black carbon-containing medium. After removing the chorion with bleach, we took photos of the eggs using a Leica MZ FLIII stereomicroscope and a QImaging MicroPublisher 5.0 RTV camera. Egg lengths were measured manually using ImageJ/Fiji [69].

Cytoplasmic actin cable density was determined by the analysis of confocal images, composed of an average of 10, 0.14 µm thick Z-sections. For each genotype, actin cables were counted along the membrane of 4–5 nurse cells, density was determined in proportion to area size, which was calculated based on the length of the selected region and thickness of the optical section. An area of at least 1,000 µm² was measured for each genotype by analyzing 4–10 egg chambers. After testing for normality, pairwise comparisons were conducted using either a Mann-Whitney test or a Student's t-test, as appropriate.

Quantification of the ring canal actin defects in nurse cells was based on the number of actin bundles associated with individual ring canals. Ring canals showing a pronounced reduction in the number of these actin bundles were classified as defective (reduced actin bundles). In most cases, such defects were associated with nuclear occlusion of the ring canal during nurse cell contraction. The frequency of defective ring canals was quantified. Length of the RC cables was determined on confocal images, by measuring length of the longest bundles manually using ImageJ/Fiji. Only appropriately aligned RCs were selected for measurements, i.e., when the actin bundles run parallel to plane of the image, RCs of at least 12 egg chambers were analyzed for all genotypes. Nuclear clogging of the ring canals was quantified manually on confocal images, stained for actin and DAPI.

All confocal imaging was performed using a Zeiss LSM800 confocal microscope with 63×/NA 1.4 oil or 40×/NA 1.3 objectives. Images were restored using Huygens Professional (Version 24.10.; Scientific Volume Imaging B.V., Hilversum, The Netherlands) and ImageJ/Fiji software. To quantify the ring canal actin defects observed upon Ena depletion, we analyzed the ring canals of 25 egg chambers of stage 10B from three independent staining.

**Intensity measurements**

Ena, Frl and Hts enrichment at the nurse cell membrane or ring canals were quantified by measuring fluorescence intensity in confocal images acquired under identical imaging and staining conditions. Intensity profiles were taken perpendicular to the membrane or ring canals in Fiji, aligned to the peak signal value, normalized, and averaged. Statistical analyses were performed using the mean peak intensity values from independent egg chambers. After testing for normality, pairwise comparisons were conducted using either a Mann-Whitney test or a Student's t-test, as appropriate.

## Western blot

The females were collected 0–8 hours after hatching and fed with yeast to promote egg production. Their ovaries were dissected two days later in ice-cold PBS. For lysis, 20 egg chambers (of stage 10B) were isolated per sample and lysed for 1 hour with lysis buffer (0.1% SDS, 0.2% NaDoc; 0.05% NP40, 150 mM NaCl, and 50 mM Tris-HCl). SDS-PAGE was performed according to standard protocols. After blotting, PVDF membranes (Millipore) were blocked in TBST with 5% dry milk powder for 1 hour at room temperature. The primary antibodies used were rat anti-FRL (1:1000; Toth et al., 2022), mouse anti-Ena (1:100; DSHB 5G2), and anti-tubulin (1:20000; DM1A, Merck KGaA). The secondary antibodies were α-rat-HRPO (1:5000) (Jackson ImmunoResearch) and α-mouse-HRPO (1:5000) (DAKO). Chemiluminescent detection was performed using the Millipore Immobilon kit. Western blot chemiluminescent signals were captured using the Alliance Q9 imaging system (UVITEC) and analyzed with Alliance (UVITEC) software. Ena signal intensities were normalized to the corresponding tubulin signal intensities.

## Supporting information

**S1 Fig. The lack of FRL abolish the actin cables in all cytoplasmic actin subpopulations.**
(TIF)

**S2 Fig. The formation and size of the ring canals is not impaired by *frl* mutants.**
(TIF)

**S3 Fig. The effect of formin knockdown on nurse cell nucleus positioning actin and egg laying.**
(TIF)

**S4 Fig. The knockdown of RhoA alters nurse cell actin organization and ring canal shape.**
(TIF)

**S5 Fig. The knockdown of Cdc42 affects FRL and Ena levels.**
(TIF)

**S6 Fig. Germline clone analysis of *ena*23.**
(TIF)

**S7 Fig. The analysis of FRL and Ena expression during oogenesis.**
(TIF)

**S8 Fig. Ena protein distribution in *frl*59 mutant egg chambers.**
(TIF)

**S9 Fig. Distribution pattern of the Ena-Flag protein upon overexpression.**
(TIF)

**S10 Fig. The distribution pattern of FRL in the nurse cells is not altered by Ena depletion.**
(TIF)

**S1 Data. Source data.**
(ZIP)

## Acknowledgments

We thank the Developmental Studies Hybridoma Bank and the Bloomington *Drosophila* Stock Center for antibodies and fly stocks. We are grateful to Gábor Csordás for critical reading and helpful comments on this manuscript. We thank for the help of Gabriella Gazsó-Gerhát, and Anikó Berente, Ildikó Velkeyné Krausz, Anna Rehák and Dorottya Csendes for technical assistance.

## Author contributions

**Conceptualization:** Rita Gombos, József Mihály.

**Funding acquisition:** József Mihály.

**Investigation:** Rita Gombos, David Farkas, Balazs Vedelek, Szilard Szikora.

**Methodology:** David Farkas, Balazs Vedelek.

**Project administration:** József Mihály.

**Software:** David Farkas.

**Supervision:** Rita Gombos, József Mihály.

**Visualization:** Rita Gombos, David Farkas.

**Writing – original draft:** Rita Gombos, József Mihály.

**Writing – review & editing:** Rita Gombos, Szilard Szikora, József Mihály.

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
