## [Decision Letter · Decision Letter 0]

12 Aug 2025

PGENETICS-D-25-00632

Spatially distinct FRL and Ena dependent actin networks coordinate nuclear positioning in Drosophila nurse cells

PLOS Genetics

Dear Dr. Mihály,

Thank you for submitting your manuscript to PLOS Genetics. After careful consideration, we feel that it has merit but does not fully meet PLOS Genetics's publication criteria as it currently stands. Therefore, we invite you to submit a revised version of the manuscript that addresses the points raised during the review process.

Please submit your revised manuscript within 60 days Oct 11 2025 11:59PM. If you will need more time than this to complete your revisions, please reply to this message or contact the journal office at plosgenetics@plos.org. Please include the following items when submitting your revised manuscript:

We look forward to receiving your revised manuscript.

Kind regards,

Ken M. Cadigan, PhD

Academic Editor

PLOS Genetics

Fengwei Yu

Section Editor

PLOS Genetics

Aimée Dudley

Editor-in-Chief

PLOS Genetics

Anne Goriely

Editor-in-Chief

PLOS Genetics

**Journal Requirements:**

3) Some material included in your submission may be copyrighted. According to PLOSu2019s copyright policy, authors who use figures or other material (e.g., graphics, clipart, maps) from another author or copyright holder must demonstrate or obtain permission to publish this material under the Creative Commons Attribution 4.0 International (CC BY 4.0) License used by PLOS journals. Please closely review the details of PLOSu2019s copyright requirements here: PLOS Licenses and Copyright. If you need to request permissions from a copyright holder, you may use PLOS's Copyright Content Permission form.

Potential Copyright Issues:

i) Figures 1i, and 6A. Please confirm whether you drew the images / clip-art within the figure panels by hand. If you did not draw the images, please provide (a) a link to the source of the images or icons and their license / terms of use; or (b) written permission from the copyright holder to publish the images or icons under our CC BY 4.0 license. Alternatively, you may replace the images with open source alternatives. See these open source resources you may use to replace images / clip-art:

4) We note that your Data Availability Statement is currently as follows: "All relevant data are within the manuscript and its Supporting Information files.". Please confirm at this time whether or not your submission contains all raw data required to replicate the results of your study. Authors must share the “minimal data set” for their submission. PLOS defines the minimal data set to consist of the data required to replicate all study findings reported in the article, as well as related metadata and methods (https://journals.plos.org/plosone/s/data-availability#loc-minimal-data-set-definition).

**Reviewers' comments:**

Reviewer's Responses to Questions

**Comments to the Authors:**

Reviewer #1: The control of the positioning of the nucleus is an essential issue for a number of cellular processes in various organisms. Cells reposition their nuclei through a variety of mechanisms, which often involve the actin or microtubule cytoskeleton. In this manuscript, the authors turn their attention to the fly egg chamber to study how actin cables control nuclear positioning. During oogenesis, the 15 nurse cells contract and transfer their cytoplasm to the oocyte via ring canals in a process known as 'dumping'. These cells are connected to each other and to the oocyte via these canals. Prior to this, actin cables originate from the nurse cell cortex and extend towards the nuclei, pushing them away from the ring canals to prevent obstruction. Despite various studies, the mechanisms that organise the actin cables responsible for holding the nuclei in place remain unclear.

The authors identify that, among the formins that act as actin nucleators, a less well-characterised one, FRL, is required in the nurse cells for the assembly of cytoplasmic actin cables and, importantly, for nuclear positioning. In line with these results, they also provide evidences that in the nurses cells, FRL is enriched at the plus end of the cytoplasmic actin filaments, as highlighted by the elongation factor Ena. Despite being present in the nurse cell membrane protrusion and at the ring canals, FRL is only indispensable for assembly of the membrane attached actin cables, but not for the ring canal associated ones. The authors illustrate that, despite Formin's capacity to act redundantly, this is not particularly significant in the formation of nurse cell actin cables with FRL. They illustrate the differential effects of the absence of FRL and Ena on cytoplasmic actin in nurse cells. FRL influences the assembly of actin cables from the plasma membrane, whereas Ena influences them from the ring canals.

They illustrate the differential effects of the absence of FRL and Ena on cytoplasmic actin in nurse cells. FRL influences the assembly of actin cables from the plasma membrane whereas Ena from the ring canals.

Importantly, the authors also demonstrate that simultaneous loss prevented the formation of nuclear-positioning actin subsets in nurse cells at the ring canal and plasma membrane. These important new results highlight that FRL and Ena are the main regulators of nuclear positioning and actin, but also suggest that these are the only two components that function in the process.

Overall, this is a nice characterization of the main regulators for the two populations of actin cables that hold and position the nuclei in the nurse cells. This manuscript contains an interesting collection of observations that will be of broad interest, although some parts of the story should be clarified and quantified more accurately.

Major points

1. The manuscript often suffers from a lack of quantification of results. This quantification is particularly important for the characterisation of actin filaments, which is a central issue in this manuscript. The authors could draw inspiration from the quantification methods used in other studies on the same process, such as those in Logan et al. (doi:10.1242/dev.197442 ). For example, this would allow the rescue of the frl mutant with mat-tubG4-UAS-FRL to be characterised more accurately, as the actin distribution shown in Fig. 1E for the rescue is quite different to that shown in Fig. 1A for the control. More generally the absence of quantification for the actin cables applies to all figures

Similarly, it would be valuable to quantify the positioning defects of the nucleus in NCs within the various genetic contexts described in the manuscript.

2. In the absence of an FRL, the actin cables connected to the ring canals are longer. However, quantifying the extent of these cables would improve the results (see Fig. 1). Furthermore, the ring canal in the frl59 mutant appears thicker and seems to present a brighter signal for Hts (Fig. S2). Quantification would help to address this point. It is possible that, in the absence of an FRL, the absence of Ena from the plasma membrane would trigger more actin at the ring canals and affect their morphology.

3. The authors analyse the distribution of FRL and Ena (see Fig. 2). Among the results, they show the presence of FRL-Ena puncta around the ring canals and propose that these dots are not directly connected to the ring canals, but rather to the area surrounding them. From the pictures presented in Figure 2, this is not obvious; colocalisation with a ring canal marker such as Hts and FRL would provide more conclusive evidence. In addition, it would be interesting to find out which part is connected to RCs and which part isn't.

4. Interestingly, the authors question the potential connection between GTPase regulation of actin assembly and FRL. They identified a genetic interaction between FRL and CDC42 and, based on this result, proposed that CDC42 is required for FRL activation. This remains a possibility that would be strongly supported by analysing FRL distribution in nurse cells with cdc42 RNAi or mutant clones.

5. By combining efficient Ena delocalisation with the FP4mito and frl mutant, the authors reveal differential effects of the absence of FRL and Ena on actin cables and nuclear positioning. However, their conclusions suffer from a lack of accurate quantification.

6. Importantly, the authors question the impact of FRL absence on Ena distribution. The results in Fig. 7 show that Ena distribution is affected at the plasma membrane. The labelling of Ena at the level of the ring canals also seems to be increased (Fig. 7B), which could be in line with the extension of the actin cable at the level of the ring canals observed in the absence of FRL. In this respect, it would be interesting to estimate the level of Ena at the ring canals in FRL mutants compared to controls.

Minor points:

With the figures 1 and S1, the three different subpopulations of cytoplasmic actin cables derived from nurse cells-nurse cells borders, nurse cells-follicle cells borders and nurse cells-oocyte borders are not easily distinguishable. An inset with a zoom highlighting these three actin populations would help.

Reviewer #2: Nuclear positioning by diverse cytoskeletal mechanisms is crucial for development and homeostasis of multiple cell types, from dividing yeast cells to migrating neurons in the developing mammalian brain and to muscles. In nurse cells of the Drosophila egg chamber, an array of cytoplasmic actin cables ensures nuclear positioning; lack of actin cables leads to small, infertile eggs.

Gombos et al. reveal that two actin regulators, the nucleation/elongation factor FRL and the elongation factor Ena, are required for the formation of these actin cables and thus for nuclear positioning. The authors suggest two types of actin cables for nuclear positioning in nurse cells, cytoplasmic actin cables and ring canal (RC) actin cables: in FRL mutants, cytoplasmic actin cables were not formed, but actin cables around ring canals were more prominent than in controls; inactivating the function of Ena showed the opposite phenotype, with less RC cables and (almost) normal cytoplasmic actin cables. Since both factors co-localised to plus-tips of both types of cables, but their respective lack of function had different effects, the authors suggest an unknown difference in the molecular machinery forming either type of cable. Furthermore, Gombos et al. show that FRL functioned partially redundantly with other formins during the formation of cytoplasmic actin cables and provide evidence suggesting that the small Rho GTPase Cdc42 regulates the formation of actin cables in nurse cells, potentially via FRL.

This manuscript identifies for the first time FRL as the main actin regulator that forms cytoplasmic actin cables in Drosophila nurse cells and thus answers a longstanding open question. In addition, the manuscript expands the list of cell types that require FRL and its interaction with Cdc42 and other formins. The provided data are based on sound genetic experiments. The manuscript and the conclusions would improve by (i) more concise / precise wording and interpretation of achieved data; (ii) by further elaborating on the main conclusion that nuclear positioning in the nurse cells requires both types of actin cables (see below).

1. Major issues:

1.1. The authors claim “nuclear positioning in the nurse cells requires the coordinated action of two spatially distinct actin networks” (e.g. lines 29-30).

1.1.1. I agree that the authors identified two actin structures, cytoplasmic cables and RC baskets/cables, that are somehow temporal and spatial distinct. But I am less convinced of both networks being required for nuclear positioning, more specifically of RC cables being required for nuclear positioning. An alternative explanation for the frl mutant phenotype could be an “out-of-balance” situation, in which the increased amount of unused Ena and/or other actin regulating proteins in frl[59] mutant nurse cells leads to an over-elongation of actin filaments forming actin baskets around RCs. These RC cables, by accident or not, are sufficient for some – I admit – remarkable nuclear positioning. So, the question arises: is there any evidence, that RC actin cables do localise nurse cell nuclei in a wild type background? Please elaborate and provide evidence or modify claim in manuscript.

1.1.2. line 135/136: please specify in the manuscript what makes RC associated cables a “distinct population” in stage 10B nurse cells. How are the authors able to distinguish RC actin cables from the other actin cables during stage 10B? By their location or by other features?

1.2. The discovery that the inactivation of Ena by FP4-mito leads to a phenotype opposite to frl[59] mutations supports the idea of two distinct subpopulations of actin cables. The use of FP4-mito might produce stronger phenotypes than the loss of Ena function, e.g. by mis-localising other proteins binding to FP4 motifs.

1.2.1. Could the authors provide further evidence (independent of FP4-mito), that lack of Ena function effects mostly RC actin cables, e.g. by analysing ‘classic’ ena mutations (e.g. ena[23])? And could the authors clarify to the reader the nature of the ena[23] mutation and include the information when discussing the results?

1.2.2. The use of FP4-mito leads to the loss of RC baskets in early egg chambers (Fig. S1 G-I). It appears to me that also filamentous actin structures at the oocyte cortex and at the membrane between oocyte and nurse cells are reduced / gone. Could it be that FP4-mito reduces generally filamentous actin and that later, i.e. stage 10B, FRL-dependent cables arise independently of Ena (or independently of whatever is recruited by FP4-mito)? Please comment and consider in Discussion.

1.3. line 177 “the lack of redundancy”: could the authors provide evidence and comment in the manuscript on efficiency of the RNAi lines and on timing? Could it be that the KD using RNAi is not early enough to affect the formation of the RC actin basket? The different strengths of Ena-RNAi and of FP4-mito effects could indicated that the RNAi needs more time for a stronger cellular effect.

1.4. line 192, chapter “Cdc42 and FRL work together in the nurse cells”

1.4.1. The conclusion “activity of FRL is regulated by Cdc42” (line 98/99 or similarly line 192) is – I guess – based on the genetic interaction of frl[59] mutations that enhance dominantly a weaker allele of Cdc42 in respect to egg lengths. Even though the literature supports this notion, the authors could substantiate their conclusion by e.g. testing how Cdc42 affects e.g. FRL (and Ena) localisation – as potential read out(s) of FRL activity. Please include the data in the manuscript. Alternatively, the conclusion should be adequately phrased. Please adapt similarly line 214 “Cdc42 is required for FRL” and line 219.

1.4.2. Fig. 4: Please add the phenotypes of Cdc42 [2] enhancement by the frl[59] mutation to the images and Cdc42[4] to the egg lengths quantification.

1.4.3. Fig.4 D: please comment in the manuscript on the stronger phenotype of Cdc42 knock down compared to frl[59] mutations. Does KD of Cdc42 affect not only FRL but also Ena? Provide evidence.

1.5. lines 264/265 “selectively affects the accumulation and localization of Ena in the developing egg chambers”.

1.5.1. Fig.7 A, B: please indicate areas with reduced Ena and provide quantification; include images showing the localisation of Ena at plus ends of RC cables and cytoplasmic actin cables lacking FRL (with quantification).

1.5.2. In Results and Discussion, please bring together Ena and FRL localisation with their described functions: are they differentially localised / expressed in nurse cells during the time course of oogenesis? Provide supporting evidence.

1.5.3. line 270 “overexpression of Ena in frl59 mutants had no such an effect (Fig. 7E)”. Could the authors show that overexpressed Ena did indeed localise to plasma membranes? Otherwise, a rescue is not expected.

1.6. When revising the Discussion:

1.6.1. Please provide a model of the functions of FRL (nucleator & elongator) and Ena (elongator) in the different subpopulations of actin cables, including how they depend on each other or not, early vs late cables, and speculate on the functionality of remaining actin cables lacking Ena or FRL.

1.6.2. A more extensive discussion of evolutionarily conserved / non-conserved mechanisms of FRL / Ena functions for the formation of filopodia-like structures would make the relevance of the results more apparent to a general audience.

2. Minor issues

2.1. Please define the types of actin cables in the beginning and use defined terms consistently in the entire text, e.g. - just one possible example - RC cables vs. cytoplasmic cables (which are oocyte cables, nurse cell cables, and follicle cables; ref. 20 of manuscript). Define what means “membrane associated”; are not all actin cables associate with a membrane? In my opinion the term “RC actin” refers to the actin of RCs proper but not to actin cables associated with RCs (e.g. line 215, line 373; similar: line 291/292, line 394).

2.2. The graphs showing the quantifications of egg lengths (Fig.1 F, Fig.3 G, Fig.4 D &E, Fig.5 O Fig.7 E, Fig.8 E) are not intuitive to me as they show the percentage of eggs within a specific class of lengths; they do not show explicitly the lengths of eggs. Thus, claims like “produce shorter eggs” (e.g. line 231) are not ‘directly’ visible. Showing absolute lengths would be less simplified and more intuitive.

2.3. In all above-mentioned graphs, the axes need to be labelled properly as % of eggs.

2.4. For all figures: please indicate the exact stage of each egg chamber shown.

2.5. line 229 “Ena is absent from the nurse cell plasma membranes and the ring canals”. Please specify imaging conditions (include in methods): were these identical to e.g. Fig1 / Fig.2, despite Ena being strongly recruited to mitochondria?

2.6. line 238/239 and Fig.6 A: specify which actin was quantified how (include in methods).

2.7. Please indicate in methods the isoform of Ena that was used for the UAS-Ena-FLAG construct.

2.8. line 310 “retains the nucleus in (roughly) the middle of the nurse cell during dumping”. This is not correct: actin cables push nuclei to the side of nurse cells (side of squamous follicle cells). Please correct.

2.9. Define terms like “mildly” (lines 235), “largely” (lines 236), and “more frequently“ (line 240, numbers for each egg chamber?) and provide quantification.

2.10. line 295 “ena; frl double mutant combinations”. Please change to an accurate genetic description.

2.11. line 348/349: please specify used deconvolution tools (include in methods).

2.12. phrase lines 351 – 355: the first process is not a process during dumping; please correct sentence.

2.13. wording: nurse cells “feeding” the oocyte (line 67) => “provide”

2.14. wording: the “intense” transport of materials towards the oocyte (line 307, 356) => “dumping” or “fast, un-selective” / “bulk” transport

**Have all data underlying the figures and results presented in the manuscript been provided?**

Reviewer #1: Yes

Reviewer #2: Yes

PLOS authors have the option to publish the peer review history of their article (what does this mean? ). If published, this will include your full peer review and any attached files.

**Do you want your identity to be public for this peer review?** For information about this choice, including consent withdrawal, please see our Privacy Policy .

Reviewer #1: No

Reviewer #2: No

**Figure resubmission:**
---

## [Decision Letter · Decision Letter 1]

1 Feb 2026

Dear Dr Mihály,

We are pleased to inform you that your manuscript entitled "Spatially distinct FRL and Ena dependent actin networks coordinate nuclear positioning in Drosophila nurse cells" has been editorially accepted for publication in PLOS Genetics. Congratulations!

Yours sincerely,

Ken M. Cadigan, PhD

Academic Editor

PLOS Genetics

Fengwei Yu

Section Editor

PLOS Genetics

Aimée Dudley

Editor-in-Chief

PLOS Genetics

Anne Goriely

Editor-in-Chief

PLOS Genetics

BlueSky: @plos.bsky.social

Comments from the reviewers (if applicable):

Please make a strong effort to respond to all the reviewers' comments.

Reviewer's Responses to Questions

**Comments to the Authors:**

Reviewer #1: In this revised manuscript, Rita Gombos and her colleagues have significantly improved their manuscript. They have performed new experiments to address several points raised by the reviewers. The authors have extensively modified the figures by integrating new results and providing accurate quantifications. They have also refined the presentation to highlight their significance.

In general, the authors have addressed and responded satisfactorily to the various points raised, clearly explaining where points could not be addressed due to experimental limitations. They have also provided detailed information on the comments and questions raised by the reviewers.

The manuscript has improved since the last submission and I strongly recommend its publication in PLOS Genetics.

Reviewer #2: The authors have resolved all issues from my first review and thereby improved significantly the manuscript.

I would like to mention one point where I have a different view than the authors:

For me the data of FP4mito and ena mutants suggest that FP4mito affects more than Enabled. Could it be, that FP4mito recruits FRL in Figure S10 panel (B)? On the other side, FP4mito is a commonly used and accepted tool for eliminating the function of Enabled; thus, I comprehend the interpretations of the authors. Furthermore, the ena mutant analyses and localisations support the model of two distinct types of cables, cytoplasmic cables and RC cables. The difference between both views lies in the strengths of phenotypes and thus in the contribution of both cable subpopulations during nuclear positioning.

**Have all data underlying the figures and results presented in the manuscript been provided?**

Reviewer #1: Yes

Reviewer #2: Yes

PLOS authors have the option to publish the peer review history of their article (what does this mean? ). If published, this will include your full peer review and any attached files.

**Do you want your identity to be public for this peer review?** For information about this choice, including consent withdrawal, please see our Privacy Policy .

Reviewer #1: No

Reviewer #2: No

**Data Deposition**

http://datadryad.org/submit?journalID=pgenetics&manu=PGENETICS-D-25-00632R1

**Press Queries**

---

## [Editor Report · Acceptance letter]

PGENETICS-D-25-00632R1

Spatially distinct FRL and Ena dependent actin networks coordinate nuclear positioning in Drosophila nurse cells

Dear Dr Mihály,

We are pleased to inform you that your manuscript entitled "Spatially distinct FRL and Ena dependent actin networks coordinate nuclear positioning in Drosophila nurse cells" has been formally accepted for publication in PLOS Genetics! Your manuscript is now with our production department and you will be notified of the publication date in due course.

With kind regards,

Zsofia Freund

PLOS Genetics

On behalf of:
